Numerical and biomass growth study of Bulimulus bonariensis (Rafinesque, 1833) (Gastropoda: Bulimulidae) under laboratory conditions

Díaz Ana Carolina anacdiaz@fcnym.unlp.edu.ar anacdy@yahoo.com.ar 1 2
Martin Stella Maris 1 3
1 División Zoología Invertebrados, Universidad Nacional de La Plata , La Plata , Buenos Aires , Argentina
2 CONICET-Consejo Nacional de Investigaciones Científicas y Técnicas , La Plata , Buenos Aires , Argentina
3 Comisión de Investigaciones Científicas , La Plata , Buenos Aires , Argentina
Banaszak Anastazia
Electronic publication date: 2024 Jan 25
Publication date: 2024
Volume: 12
Electronic Location ID: e16803
Received 2023 Oct 2; Accepted 2023 Dec 27
Copyright: ©2024 Díaz and Martin
Copyright year: 2024
Copyright holder: Díaz and Martin
License: This is an open access article distributed under the terms of the Creative Commons Attribution License, which permits unrestricted use, distribution, reproduction and adaptation in any medium and for any purpose provided that it is properly attributed. For attribution, the original author(s), title, publication source (PeerJ) and either DOI or URL of the article must be cited.
License URL: https://creativecommons.org/licenses/by/4.0/

Keywords: Clutch parameters, Growth models, Survival, Mortality, Life expectancy

Funding: Facultad de Ciencias Naturales y Museo, Universidad Nacional de La Plata (Proyect N870) CONICET which awarded the Internal Doctoral Fellowship during 2016–2022 Financial support for this work was provided by an institutional project from the Facultad de Ciencias Naturales y Museo, Universidad Nacional de La Plata (Proyect N870) and from CONICET which awarded the Internal Doctoral Fellowship during 2016–2022. The funders had no role in study design, data collection and analysis, decision to publish, or preparation of the manuscript.

==============================
Bulimulus bonariensis is considered a species of relevance to agribusiness, having been declared a pest with indirect damage because of its negative effects on several crops such as soybeans, chickpeas, and corn in central and northern Argentina. The objective of this work was to analyze the growth pattern of a population born under laboratory conditions, to explore population aspects such as survival and mortality, to estimate the age and size at gonadal maturity and first reproduction, and to contribute to the knowledge of the reproductive biology of this gastropod. From the clutches obtained, the basic biologic parameters were calculated and the individuals hatched under laboratory conditions counted and measured every two weeks. The clutches contained an average of 44 eggs, which took about 13.7 days to hatch at a birth rate of 41.82%. The growth pattern in the five clutches was analyzed individually, and the logistic model used was the one with the highest degree of fit to that observed growth pattern, followed by the Gompertz model, and finally the von Bertalanffy model. In addition, the models were applied to the 102 specimens analyzed together as a cohort, where the best fitting model was also proved to be the logistic growth model. A concave type III survival curve was obtained from the horizontal life table. The cohort was reduced by 48% during the first 50 days after birth. Beyond one month of hatching, life expectancy gradually increased and remained high between 65–302 days of life. After day 330, life expectancy decreased and only 13.72% exceeded one year of birth, with an average length of 16.68 mm. The last specimen died after 23 months at a total length of 20.24 mm, and the life expectancy was estimated at almost three years. In addition, it was inferred that gonadal maturity, when these gastropods reach 12 mm of total shell length, is reached after 200 days of life. Therefore, the individuals that are born are able to reproduce for the first time a year after birth, when they have the approximate size of 16.68 mm.

Introduction

The Bulimulidae family comprises species native to the tropics and subtropics of South America (Salvador et al., 2023). With 26 genera, this family of terrestrial gastropods is very diverse. Bulimulids can be found associated with a wide variety of environments, such as herbaceous vegetation, dry leaf litter, rocky walls, and arboreal habits; with an altitudinal distribution range from sea level to 5,200 m (Breure, 1979). The family is highly adaptable to a wide variety of climatic conditions and habitats from near-desert to temperate humid forest climates (Coppois, 1995). In addition, these snails have the ability to colonize new areas and a wide variety of ecologic niches with different types of vegetation, temperature, and humidity conditions (Cabrera et al., 2021). In particular, the genus Bulimulus has a native neotropical distribution (Breure, 1979). In relation to ecological and life history research, the only few available publications from South America involve studies on the life cycles, growth, and reproduction of Bulimulus tenuissimus (D’Orbigny, 1835) from Brazil (Meireles et al., 2008; Meireles et al., 2010; Silva et al., 2008; Silva et al., 2009; Silva et al., 2013). There are no equivalent studies from Argentina on other species of Bulimulus; with most of the Argentine work having taxonomic, not biological.

Since the latter half of the 20th century, in temperate and tropical regions, the occurrence of gastropod crop pests has increased, owing to the production of new crops and the intensification of production systems. In addition, regional and international trade has favored the spread of species, where by exotic species have been introduced that have become serious agricultural pests (Barker, 2002; Raut & Barker, 2002; Robinson, 1999). Moreover certain some native gastropod species have been favored in their survival and reproduction as possible consequences of a greater food supply through enhanced crop production, or other effects of habitat modifications. This increase in agricultural production, however, causes a detriment to natural environments, eliminating natural biologic controllers (Barker, 2002). The impact of terrestrial gastropods on agricultural production can be direct, causing damage to some part of the plant and thus affecting its growth; or indirect such as clogging irrigation systems, drip lines, and harvesting machines. The damage to agricultural production becomes translated into a decrease in yield and product quality (Barker, 2002; Frana & Massoni, 2011; Matamoros Torres, 2011; Virgillito et al., 2015).

In Argentina, Frana & Massoni (2007); Frana & Massoni (2011) reported the presence of Bulimulus bonariensis (Rafinesque, 1833) in soybean crops and declared the species dominant in the province of Santa Fe; the authors even carried out exploratory control treatments. During grain harvesting, the snails clogged the sieves, causing crop loss and hours of cleaning work. This same problem during harvesting was repeated with chickpea crops in Córdoba, but the snails also affected the appearance of the grain. Therefore Peralta (2016) catalogued this species as a pest causing indirect damage. Snails of the genus were also found on corn crops in Cañada de Luque, Córdoba, on the basal leaves of the crop in the phenologic stage V4 (Rumi, 2016, personal communication). In addition, producers of yerba mate (Ilex paraguariensis) in Misiones reported that a population of Bulimulus had invaded plantations rendering the leaf nonviable for its use (Noticias La Región, 2013). Agricultural magazines expressed concern about the effect on different crops, where the land snails cause great damage in a very short time (e. g., Campo Agropecuario, 2021). Because they can consume up to 50% of their weight in one night, causing serious damage to leaves, petioles, and shoots; that destruction which reduces photosynthetic activity, affecting the growth, yield and quality of the crop, and can even lead to the total loss of the plant (Vaca, 2021).

In view of these considerations, the objective of the work to be described here was to monitor a population of B. bonariensis born under laboratory conditions, analyze the growth pattern, and evaluate the best-fit model. At the same time, we intended to explore basic aspects of the population such as survival and mortality, and to estimate the age and size at first reproduction. It is hoped that the knowledge generated from these studies would constitute an initial necessary and indispensable step in more fully understanding the general biology and especially the growth of B. bonariensis. Therefore, our aim was to provide information that will enable the development of protocols or actions in order to eventually mitigate the negative impacts of these gastropod pests through prevention and agronomic control that will be useful for national enforcement agencies and of practical advantage in the field.

Materials & Methods

Collection and maintenance of snails in the laboratory

The present study was initiated with a stock of 10 adult individuals of B. bonariensis available at the laboratory SERByDE–Unidad de Servicios en Bioensayos y Diagnósticos Ecotoxicológicos (División Zoología Invertebrados, Museo de La Plata, UNLP). The snails were maintained under controlled laboratory conditions (photoperiod programmed at 12 h of artificial light, 12 h of darkness; 75–85% humidity; and 23–25 °C), in a glass terrarium (15 × 15 × 10 cm), conditioned to recreate their natural habitat, with a soil substrate and a layer of dry leaves and vegetation. The substrate was sterilized at 120 °C for 1 h in a drying oven (Silva et al., 2009). The adults and their offspring were fed ad libitum with a mixed diet (Meireles et al., 2008) consisting of lettuce (Lactuca sativa Linné, 1758) and a balanced feed prepared specifically for terrestrial gastropods, according to the nutritional composition suggested by Barbado (2003): i. e., wheat flour 26%, corn flour 26.5%, soybean flour 15%, bone meal 15%, calcium carbonate 15%, salt 10.5%, vitamins and minerals 2%.

The adult individuals were checked daily and, when clutches were observed, they were separated from the parent to be kept in smaller terraria (10 × 10 × 7 cm). At hatching, the juveniles were measured and maintained under the conditions described above. Upon reaching 6 mm of total shell length, each specimen was kept in an individual terrarium (10 × 10 × 7 cm). The experiment was performed in accordance with the recommendations from Comité Institucional para el Cuidado y Uso de Animales de Estudio (CICUAE; FCNyM-UNLP; approval number 009.06.2023).

Biologic parameters of clutches

The following parameters of the clutches were obtained following Daguzan et al. (1981) and which are described in Díaz (2022): fecundity coefficient, hatching coefficient, percent birth rate, embryonic mortality rate, incubation duration.

Growth models

The total shell length was measured from the day the snails were born (day zero), and the growth was studied until the last animal died. The measurements were made every two weeks up to a size of 6 mm (Meireles et al., 2008) under a Leica EZ4 stereo microscope with a micrometer eyepiece; the individuals larger than 6 mm were measured with a millimeter vernier caliper (precision of 0.01 mm).

The data of the modal length in mm calculated over time revealed a growth pattern that was evaluated under different models in order to determine which of the latter most accurately explained the individual growth of the population.

The models evaluates were the following:

–von Bertalanffy’s model (von Bertalanffy, 1938), described by the following formula: Lt=Lmax1−e−k(t−to).

Because of the large number of gastropod species, many of which were terrestrial, that have evidenced a growth pattern with a high degree of fit to this model, it was the first one tested.

–The logistic model: the growth pattern observed was also adjusted to this model, the equation being: Lt =Lmax/1 + e(b−kt) (Brody, 1945).

–Finally, was evaluated the degree of adjustment to the Gompertz model (Gompertz, 1825), Lt = Lmax*e-e−k(t−t0).

The parameters are described as: Lt = modal length at time t (mm); Lmax = maximum asymptotic length (mm); e = base of the natural logarithm; b = phase with smaller values before starting the slope; k = growth constant; t = time; to = hypothetical time when the snail length equals “zero”.

In the von Bertalanffy model, the parameter maximum asymptotic length parameter was obtained by applying Walford’s method (Walford, 1946).

For the calculation of the parameters of the Logistic and Gompertz models, we started from the estimates for the von Bertalanffy model. As realized by the SPSS Statistics program, using the Levenberg-Marquadt algorithm provided the parameters through an iterative process. The theoretical curves were contrasted with the observed values. Non-parametric tests (chi2 and Wilcoxon test) were applied with the SPSS Statistic program, where the degrees of freedom correspond to the number of Lt-1. In addition, the slope of the line between the observed and expected values (b), the correlation coefficient (r), and the coefficient of determination (r2) were calculated by means of linear regression analyses. Two statistical criteria were used to select the model with the best fit to the observed growth pattern. First, the model with the minimum negative log-likelihood (LL) value was identified, and then, as a second statistical criterion, the Akaike Information Criterion (AIC) values were obtained and compared, where a lower value indicates a more adequate and parsimonious fit to the adjusted model (Averbuj, Escati-Peñaloza & Penchaszadeh, 2015; Haddon, Mundy & Tarbath, 2008; Helidoniotis et al., 2011).

In order to define which of the models most accurately explained the individual growth of B. bonariensis, an analysis of each group of individuals, each belonging to a specific clutch, was carried out. Then, to verify if there were significant differences existed between the models, applied to the groups/clutches, a one-way analysis of variance (ANOVA) was performed on the parameters maximum asymptotic length and growth constant. Subsequently, an integrated analysis of the individuals from all the clutches was carried out, with all the clutches as a single cohort.

Finally, Munro’s Phi index was applied, where Φ =Log(k)+2Log(lmax) (Pauly & Munro, 1984), in order to compare the growth capability of B. bonariensis with that of four exotic species present in Argentina: Allopeas gracile, Lissachatina fulica, Cornu aspersum and Rumina decollata.

Life table

A horizontal life table of the snails born in the laboratory was drawn up (Rabinovich, 1980; Smith & Smith, 2007). The day in which individuals were born was considered as day zero, and the number of survivors was grouped into age classes; every nine days during the first month and thereafter every 14 days. Thus, the values of survival, average survival in age intervals, mortality, mortality rate, and life expectancy were recorded, where:

x = age in days.

n(x) = number of individuals alive at age (x).

l(x) = (nx/n0) number of individuals surviving to age (x).

d(x) = (n(x+1)-n(x)) corresponds to mortality. The number of deaths between age (x) and age (x + 1).

q(x) = (d(x)/n(x)) is the age-specific mortality rate (x).

L(x) = (n(x)+n(x+1)/2) average number of individuals alive during the age interval (x) to (x + 1).

T(x) = ∑x L(x) corresponds to the total days left to live or projected life expectation of survivors who have reached age (x).

e(x) = T(x)/n(x) is the life expectancy at age (x).

Results

Biologic parameters of clutches

The adult specimens oviposited seven times, providing a total of 125 newly hatched individuals. The clutches (Fig. 1A) contained of an average of 44 eggs (SD = 12.6), where the minimum number of eggs was 21 and the maximum was 59. The eggs were spherical and gelatinous, with calcium carbonate crystals, and whitish in color with an average diameter of approximately 1.7 mm (Figs. 1A–1B). A few days before hatching, the eggs turned a brownish color, evidencing the embryonic shell. The development time until hatching was between 10–17 days, at an average of 13.7 days (SD = 2.3 days). The birth rate, the proportion of hatching juveniles, was 41.82% on average (SD = 15.44%) (Table 1) (Fig. 1C).

Figure 1 Clutch and juvenile Bulimulus bonariensis.

(A) One of the clutches studied. (B) Detail of the egg. (C) Recently hatched juvenile. Scale A-C = 1 mm. Photo credit: Ana Carolina Díaz.

Table 1 Biologic parameters of Bulimulus bonariensis clutches.

Fecundity coefficient (no. eggs/clutch), hatching coefficient (no. young hatched), birth rate (no. juveniles hatched/no. eggs in the clutch x 100), embryonic mortality rate (no. aborted embryos/no. eggs in the clutch x 100), incubation duration (days until hatching).

Clutch	Fecundity coefficient	Hatching coefficient	Birth rate	Embryonic mortality rate	Incubation duration (days)	
1	32	17	53.13	46.87	12	
2	44	20	45.45	54.55	10	
3	50	28	56	44	14	
4	48	29	60.42	39.58	12	
5	21	8	38.1	61.9	15	
6	57	11	19.3	80.7	17	
7	59	12	20.34	79.66	16	

Growth models

Snail growth was studied for 23 months, until the last individual died, with no offspring being produced during this period. Upon monitoring the snails, the growth analysis was applied to only those corresponding to the first five clutches; the two subsequent clutches died very shortly after hatching and for this reason were not included in the analysis.

Analysis of growth in the groups/clutches individual

Clutch/group 1

From this clutch, 17 individuals were born, with an average initial length of 2.31 mm, and lived to 1.5 years, with a maximum modal length of 20.68 mm. The growth pattern (Fig. 2A) exhibited a satisfactory fit to the three models (39 g.l. and p = 0.05) for which the following chi2 values were calculated: 6.34 (von Bertalanffy); 3.06 (Logistic); 2.53 (Gompertz). Furthermore, no significant differences were found among the models by the Wilcoxon test: Z =−0.43, p = 0.667 (von Bertalanffy); Z = 0, p = 1 (Logistic); Z =−0.094, p = 0.925 (Gompertz).

Figure 2 Observed and estimated growth curves for each clutch/group of Bulimulus bonariensis.

(A) Growth curves of clutch/group 1. (B) Growth curves of clutch/group 2. (C) Growth curves of clutch/group 3. (D) Growth curves of clutch/group 4. (E) Growth curves of clutch/group 5. The pink dotted curve represents the modal-length values observed in each group with the corresponding standard deviations (black dots) lying above and below. The continuous lines correspond to the estimated curves for each model: von Bertalanffy (black), Logistic (blue) and Gompertz (gray).

Clutch/group 2

In this case, the newly hatched specimens (20) had a modal length of 2.28 mm and were measured to 20.2 mm at 1.5 years of age (Fig. 2B). The fit to the models was also quite satisfactory (41 g.l. and p = 0.05), with the calculated chi2 values being: 5.63 (von Bertalanffy); 3.89 (Logistic); 3.15 (Gompertz). Moreover, according to Wilcoxon tests, no significant differences were found among the models: with Z =−0.056, p = 0.955 (von Bertalanffy); Z =−0.131, p = 0.896 (Logistic); Z =−0.006, p = 0.995 (Gompertz).

Clutch/group 3

In this group, the clutch consisted of 28 individuals that measured on average 1.8 mm at birth and reached 20.24 mm in modal shell length at almost 2 years of age (Fig. 2C). The fit (50 g.l. and p = 0.05) was quite satisfactory, with the calculated chi2 values being: 9.11 (von Bertalanffy); 2.2 (Logistic); 1.25 (Gompertz). According to Wilcoxon tests, no significant differences were found among the models: Z =−0.572, p = 0.567 (von Bertalanffy); Z =−0.178, p = 0.859 (Logistic); Z = 0, p = 1 (Gompertz).

Clutch/group 4

The clutch initially contained 29 neonate specimens measuring 1.88 mm in modal length. The juveniles were measured until the last specimen reached a length of 19 mm, after a little over 1.5 years of life (Fig. 2D). The degree of fit to the models (44 g.l. and p = 0.05) counted with the calculated chi2 values: 4.27 (von Bertalanffy); 2.38 (Logistic); 2.23 (Gompertz). According to Wilcoxon tests, no significant differences were found among the models: Z =−0.926, p = 0.355 (von Bertalanffy); Z =−0.006, p = 0.995 (Logistic); Z =−0.497, p = 0.619 (Gompertz).

Clutch/group 5

From this clutch, eight individuals were born with a modal length of 1.74 mm and were measured to a modal length of 20.19 mm at 1.5 years of life (Fig. 2E). The degree of fit to the models (39 g.l. and p = 0.05) had the calculated chi2 values: 15.26 (von Bertalanffy); 3.87 (Logistic); 3.96 (Gompertz). According to Wilcoxon tests no significant differences were found among the models: Z =−0.108, p = 0.914 (von Bertalanffy); Z =−0.188, p = 0.851 (Logistic); Z =−0.121, p = 0.904 (Gompertz).

Table 2 lists the slope of the line (b), the correlation coefficient (r), the coefficient of determination (r2), and the values of both information criteria (LL) and (AIC) together with the corresponding model formula for each group.

In all five groups, the model with the highest degree of fit to the observed growth pattern was the logistic model, manifesting in all cases the highest correlation and determination coefficient together with the lowest values of both log likelihood and Akaike’s information criteria. After the logictic model, the Gompertz model was next in the degree of correspondence followed by the von Bertalanffy model.

In contrast, among the values of maximum asymptotic length calculated for each model in the five groups, significant differences were found between them by one-way ANOVA F (2, 12) =59.77 p < 0.05. In the Tukey post-hoc test, a significant difference was found in the maximum asymptotic length calculated for the von Bertalanffy model relative to the logistic model q (p < 0.05, 3, 2) =14.94 and compared to the Gompertz model q (p < 0.05, 3, 2) =10.93. No significant differences were found between values for the maximum asymptotic length in the logistic and Gompertz models q (p < 0.05, 3, 2) = 4.004.

Finally, when an ANOVA was performed between the growth constants of the five groups for the three models; no significant differences were found F (4, 10) =0.088 p = 0.983 and the homogeneity of variances was fulfilled (Levene F (4, 991) = 0.048 p = 0.994).

Table 2 Statistics, information criteria and equations for the models applied in each group of Bulimulus bonariensis.

	Model	b	r	r2	LL	AIC	Equation	
Group 1	von Bertalanffy	0.973	0.986	0.973	−169.2	347	Lt = 32.664(1-e−0.762(t−0.81))	
Logistic	0.980	0.99	0.983	−164.1	336.7	Lt = 22.41/1+e(4.992−3.537∗t)	
Gompertz	0.984	0.99	0.981	−168.7	345.9	Lt = 24.472*e −e−2.708(t−2.123)	
Group 2	von Bertalanffy	0.989	0.987	0.975	−178.4	365.2	Lt = 32.35(1-e−0.665(t−0.82))	
Logistic	0.977	0.99	0.98	−173.8	356.1	Lt = 22.475/1+e(4.318−2.851∗t)	
Gompertz	0.98	0.99	0.98	−176.5	361.4	Lt = 25.207*e −e−2.275(t−1.66)	
Group 3	von Bertalanffy	0.982	0.994	0.987	−214.1	436.5	Lt = 31.065(1-e−0.586(t−0.86))	
Logistic	0.984	0.996	0.992	−204.5	417.3	Lt = 20.919/1+e(4.554−2.817∗t)	
Gompertz	0.992	0.996	0.993	−209.3	426.9	Lt = 26.261*e −e−2.499(t−1.73)	
Group 4	von Bertalanffy	0.991	0.99	0.98	−185.2	378.9	Lt = 30.094(1-e−0.596(t−0.87))	
Logistic	0.989	0.995	0.99	−181.4	371.3	Lt = 20.198/1+e(4.382−2.733∗t)	
Gompertz	0.991	0.995	0.989	−183.7	375.8	Lt = 26.226*e−e−2.309(t−1.588)	
Group 5	von Bertalanffy	0.994	0.979	0.959	−169.7	347.9	Lt = 30.324(1-e−0.795(t−0.968))	
Logistic	0.986	0.99	0.98	−166.4	341.4	Lt = 23.055/1+e(4.727−2.898∗t)	
Gompertz	0.98	0.988	0.975	−168.4	345.3	Lt = 26.865*e −e−2.392(t−1.58)	

Growth analysis for all the individuals as a cohort

All five clutches hatched within a few days of each other (every 8–9 days) and this occurred over the course of 52 days. For this reason, all five groups were considered a cohort, and the day they hatched was unified as time zero. Thus, the total shell length data were pooled and modal lengths were calculated in order to test the degree of fit to the different models.

von Bertalanffy’s model

Walford’s method was used to calculate the maximum asymptotic length (Fig. 3), which was 26.849 mm, while the maximum observed length was 20.73 mm.

Figure 3 Walford graphical method upon considering the groups 1–5 of Bulimulus bonariensis as a cohort.

The unified growth curve evidenced a satisfactory fit to the von Bertalanffy model (Fig. 4A) which at 50 g.l. and p = 0.05 had a chi2 value = 3.6. No significant differences were found between the observed and theoretical modal lengths according to the Wilcoxon test at a value of Z =−0.703 and p = 0.482. Table 3 list the slope of the line between the observed and expected values (b), the correlation coefficient (r) and the coefficient of determination (r2), together with the values of both the log-likelihood (LL) and the Akaike Information Criterion (AIC) in comparison to the other models. The von Bertalanffy’s model was reflected in the following equation: Lt=26.8491−e−0.784t−0.749.

Figure 4 Growth curve of the Bulimulus bonariensis cohort adjusted to each model.

The dotted line (pink) corresponds to the modal length values with their corresponding ± standard deviation (black dots) and the solid line (black) corresponds to the estimated curve for each model: (A) von Bertalanffy, (B) logistic, (C) Gompertz.

Logistic model

The degree of fit of the observed growth pattern to this model was quite satisfactory (Fig. 4B) at 50 g.l. and p = 0.05 with a chi2 value = 1.83. No significant differences were found between the observed and theoretical modal lengths according to the Wilcoxon test at a value of Z = −0.037 and p = 0.97. Table 3 lists the slope of the line between the observed and expected values (b), the correlation coefficient (r), and the coefficient of determination (r2), together with the values of both the log-likelihood (LL) and the Akaike Information Criterion (AIC), in comparison to the other models. The growth equation with the estimated parameters for this model is: Lt=20.439/1+e1.915−0.009t.

Gompertz model

The observed growth pattern also evidenced a satisfactory fit to this model (Fig. 4C) at 50 g.l. and p = 0.05 with a chi2 value =1.78. No significant differences were found between the observed and theoretical modal lengths according to the Wilcoxon test at a value of Z =−0.309 and p = 0.757. Table 3 lists the slope of the line between the observed and expected values (b), the correlation coefficient (r), and the coefficient of determination (r2), together with the values of both the log-likelihood (LL) and the Akaike Information Criterion (AIC), in comparison with the other models. The growth equation with the estimated parameters for this model is: Lt=21.462∗e−e−2.673t−2.121.

The three models could be fitted to the growth pattern of the cohort. As in the case of the groups, when analyzed individually, the model with the best fit to the curve studied was the logistic growth model. In addition, the growth parameters of B. bonariensis were compared to the corresponding values from four exotic species: A. gracile, L. fulica, R. decollata, or C. aspersum. Table 4 summarizes the parameters maximum asymptotic length (Lmax), the growth constant (k) and Munro’s index (Φ), calculated for each model in the four exotic species in relation to the native species studied as evaluated by the von Bertalanffy other logistic model.

Table 3 Statistics and information criteria for the growth models applied to the cohort of Bulimulus bonariensis studied.

Model	b	r	r2	LL	AIC	
von Bertalanffy	0.932	0.988	0.976	−205.2	418.8	
Logistic	0.992	0.996	0.993	−193.4	395.1	
Gompertz	0.989	0.994	0.989	−199.5	407.3	

Table 4 Comparison of parameters of Bulimulus bonariensis with alien species.

Species	Lmax	k	Φ	Model	Reference	
Lissachatina fulica	113.3	0.75	3.98	von Bertalanffy	Carvalho da Silva & Omena (2014)	
Rumina decollata	29	5.54	3.66	von Bertalanffy	Values calculated from Table IV Hines (1951)	
Allopeas gracile	6.756	0.286	1.11	von Bertalanffy	Nandy & Aditya (2022)	
Bulimulus bonariensis	21.46	0.78	2.55	von Bertalanffy	Present study	
Cornu aspersum	40	2.2	3.66	logistic	Nicolai et al. (2010)	
Bulimulus bonariensis	20.43	1.9	2.9	logistic	Present study	

Life table

B. bonariensis (n0 =102) exhibited a concave Type III survival curve (Rabinovich, 1980) (Fig. 5A) (Table 5). During the first 50 days of follow-up a 48% reduction in the population was recorded with and only 13.72% reaching one year of life at an average length of 16.68 mm. The last specimen died after 23 months at a total length of 20.24 mm. Since in natural populations a maximum size of 29.51 mm has been recorded, we can anticipate that specimens of this species could live up to 3 years, with a life expectancy calculated in the life table (Table 5) of 1,020 days or 2.79 years. The maximum peak mortality (dx) was recorded before one month of life (day 24) (Fig. 5B), the mortality rate (qx) up to day 528 averaged 0.06 (0−0.21), and two peaks were subsequently recorded on days 540 and 610 respectively (Fig. 5C). Upon survival of the month of hatching, the life expectancy began to increase (Fig. 5D) and remained high between 65–302 days of life, exhibiting two peaks of 18 on days 176 and 261, respectively. After day 330 life expectancy decreased.

Figure 5 Survival, mortality and life expectancy curves of the cohort corresponding to Bulimulus bonariensis.

(A) Survival curve. (B) Mortality. (C) Mortality rate. (D) Life expectancy.

Table 5 Horizontal life table of Bulimulus bonariensis.

x	n(x)	l(x)	d(x)	q(x)	L(x)	T(x)	e(x)	
0	102	1.00	9	0.09	97.5	1020.0	10.00	
9	93	0.91	11	0.12	87.5	922.5	9.92	
24	82	0.80	15	0.18	74.5	835.0	10.18	
35	67	0.66	14	0.21	60.0	760.5	11.35	
51	53	0.52	9	0.17	48.5	700.5	13.22	
65	44	0.43	4	0.09	42.0	652.0	14.82	
79	40	0.39	2	0.05	39.0	610.0	15.25	
92	38	0.37	5	0.13	35.5	571.0	15.03	
105	33	0.32	2	0.06	32.0	535.5	16.23	
119	31	0.30	1	0.03	30.5	503.5	16.24	
134	30	0.29	3	0.10	28.5	473.0	15.77	
147	27	0.26	3	0.11	25.5	444.5	16.46	
161	24	0.24	3	0.13	22.5	419.0	17.46	
176	21	0.21	0	0.00	21.0	396.5	18.88	
191	21	0.21	1	0.05	20.5	375.5	17.88	
204	20	0.20	0	0.00	20.0	355.0	17.75	
218	20	0.20	1	0.05	19.5	335.0	16.75	
232	19	0.19	1	0.05	18.5	315.5	16.61	
246	18	0.18	3	0.17	16.5	297.0	16.50	
261	15	0.15	0	0.00	15.0	280.5	18.70	
275	15	0.15	0	0.00	15.0	265.5	17.70	
288	15	0.15	0	0.00	15.0	250.5	16.70	
302	15	0.15	0	0.00	15.0	235.5	15.70	
316	15	0.15	1	0.07	14.5	220.5	14.70	
330	14	0.14	0	0.00	14.0	206.0	14.71	
344	14	0.14	0	0.00	14.0	192.0	13.71	
359	14	0.14	0	0.00	14.0	178.0	12.71	
365	14	0.14	1	0.07	13.5	164.0	11.71	
379	13	0.13	0	0.00	13.0	150.5	11.58	
393	13	0.13	1	0.08	12.5	137.5	10.58	
407	12	0.12	0	0.00	12.0	125.0	10.42	
423	12	0.12	1	0.08	11.5	113.0	9.42	
435	11	0.11	0	0.00	11.0	101.5	9.23	
449	11	0.11	0	0.00	11.0	90.5	8.23	
463	11	0.11	0	0.00	11.0	79.5	7.23	
478	11	0.11	0	0.00	11.0	68.5	6.23	
491	11	0.11	1	0.09	10.5	57.5	5.23	
512	10	0.10	1	0.10	9.5	47.0	4.70	
528	9	0.09	0	0.00	9.0	37.5	4.17	
540	9	0.09	4	0.44	7.0	28.5	3.17	
554	5	0.05	1	0.20	4.5	21.5	4.30	
568	4	0.04	1	0.25	3.5	17.0	4.25	
582	3	0.03	0	0.00	3.0	13.5	4.50	
596	3	0.03	0	0.00	3.0	10.5	3.50	
610	3	0.03	2	0.67	2.0	7.5	2.50	
624	1	0.01	0	0.00	1.0	5.5	5.50	
638	1	0.01	0	0.00	1.0	4.5	4.50	
652	1	0.01	0	0.00	1.0	3.5	3.50	
666	1	0.01	0	0.00	1.0	2.5	2.50	
682	1	0.01	0	0.00	1.0	1.5	1.50	
694	1	0.01		0.00	0.5	0.5	0.50	
Notes.

Statisticsx age represented in days

n(x) number of individuals at age x

l(x) survival

d(x) mortality

q(x) mortality rate

L(x) average survival in the age interval

T(x) life expectation

e(x) life expectancy

Discussion

An essential characteristic in the life history of terrestrial mollusks is the number of eggs per clutch, which parameter varies among species. Even within a species variations occur depending on size and/or age of the parent and the environmental factors (Heller, 2001).

In Argentina, among the exotic species such as Deroceras reticulatum, an average of three eggs has been recorded, while D. laeve was recorded as ovipositing around six eggs (Clemente et al., 2007), and in R. decollata three–13 eggs were counted with an average of eight (Álvarez González, Pizá & Cazzaniga, 2019). C. aspersum is among the exotic species with a high number of eggs whose ovipositions average 118 eggs (Daguzan et al., 1981) and L. fulica whose clutches range from 10 to 400 eggs (Armiñana García, Fimia Duarte & Iannacone, 2020). In A. gracile, Nandy & Aditya (2022) observed that the snails can increase the number of eggs per clutch and that they oviposit between 46-244 eggs during their lifetime.

Within the family Bulimulidae, studies on life cycles, growth, and reproduction are scarce; but some studies have been reported on neotropical representatives. In B. tenuissimus the average number of eggs per clutch for individuals arranged in pairs was 48.3 (Silva et al., 2008) and 35.9 when the substrate was moistened daily (Silva et al., 2009), B. bonariensis lies within that range with 44 eggs on the average. In Bostryx conspersus, Ramírez (1988) reported a lower clutch size (10–34). In other larger Orthalicoid snails, such as Placostylus, however, clutches vary between 25 and 450 eggs at an average of 205 (Brescia et al., 2008).

The average time of embryo development to hatching in B. tenuissimus is ten days more (23.62) (Silva et al., 2008) than in B. bonariensis (13.7); for B. conspersus, a longer incubation time was also observed (19.56 days) (Ramírez, 1988), and in Placostylus the interval is 22 days on the average (Brescia et al., 2008). The average birth rate in B. tenuissimus varies between 39% (Silva et al., 2008) and 53% (Silva et al., 2013), whereas that of B. bonariensis was intermediate between these values, at 41.82%. This is a characteristic that must be taken into consideration when explaining the emergence of populations in crops (Frana & Massoni, 2007; Frana & Massoni, 2011; Peralta, 2016). On the other hand, this parameter is also fundamental in comparing native and introduced species in the country such as L. fulica and R. decollata, whose birth rate is in both cases higher at between 90% and 63%, respectively (Armiñana García, Fimia Duarte & Iannacone, 2020; Álvarez González, Pizá & Cazzaniga, 2019). However, the highest birth rate was recorded for A. gracile with 99% (Nandy & Aditya, 2022). In C. aspersum, though, the natality rate was only 31.8% (Daguzan et al., 1981). In addition, the juveniles of A. gracile hatch between 1–3 days (Nandy & Aditya, 2022) and those of L. fulica at 11 days, unlike other exotic species that exhibit a longer development time, such as Deroceras reticulatum and D. leave, each with 16 days of incubation (Clemente et al., 2007), C. aspersum 22 days, and R. decollata between 25 (Hines, 1951) and 40 days (Álvarez González, Pizá & Cazzaniga, 2019).

Other principal life history traits are growth patterns, developmental time and adult length at first reproduction (Benbellil-Tafoughalt & Koene, 2015). An analysis of the groups of B. bonariensis together as a cohort indicated a quite satisfactory fit to the three models proposed, with the logistic model being the most explanatory of the observed pattern. García et al. (2006) also observed an appropiate fit to several models for C. aspersum, such in square root, exponential, linear and logistic models, the latter being the one with the highest correlation coefficient, a finding that was corroborated by Nicolai et al. (2010). In B. tenuissimus, the curves published were essentially sigmoidal although Silva et al. (2009) did not analyze that growth pattern.

The analysis of the individual groups/clutches of B. bonariensis also evidenced a good fit to the three models, with the logistic model and the Gompertz models having the best fit. Notwithstanding, upon consideration of the correlation coefficients, determination coefficients, and both information criteria; the logistic model explained, in all instances, the growth pattern more accurately.

It is important to highlight that mathematical growth models provide tools and numerical results that enable a more accurate interpretation of biologic processes, as well as obtaining parameters that facilitate comparisons. In this regard, with respect to the growth constant, no significant differences were found between the groups. The other biologically relevant parameter is the maximum asymptotic length. In the material deposited in the Malacological Collection of the La Plata Museum, it was observed that the largest specimen measures 30.22 mm in total length of the shell (MLP-Ma 2315). During the cohort growth study it was observed that the von Bertalanffy model yielded an asymptotic maximum length value of 26.84 mm; but in the logistic model, the maximum asymptotic length calculated was lower (at 20.43 mm). However, it is important to highlight that the analysis of each group allowed us to strengthen the logistic model as the most explanatory model and also indicated that the von Bertalanffy model overestimated the values of maximum asymptotic length in four of the five groups, since those estimates were greater than 30 mm, with only group 4 evidencing a value of 30 mm. The logistic model predicted the lowest asymptotic lengths, while the Gompertz model not only resulted in a good fit but also gave closer values to the real data of the maximum length parameter observed both in the field and in the collection material.

In addition, the Gompertz model implies a sigmoidal growth with three phases, an initial slow growth, a rise to an inflection point, and a further increase until reaching an asymptote. This kinetics are similar to those of the logistic model but distinct from the latter by the location of the inflection point located between 35–40% in the Gompertz model and at 50% in the logistic model (Mínguez, 2016). These similarities account for the high correlation between these two models and the observed growth patterns, which are more similar to a sigmoid curve than to a parabolic curve as in the von Bertalanffy model. From the above considerations, we can conclude that both the logistic and the Gompertz models best explain the growth pattern. The logistic model, however, was the most explanatory from the mathematical point of view, whereas the Gompertz model was the one that most precisely adjusted the maximum length parameter.

A logistic growth curve is made up of an early stage of exponential growth, a linear growth in which energy is focussed on maintenance, a decrease in growth and an asymptote at senescence (Karkach, 2006). In B. bonariensis the second phase of the curve, between days 200–300 (∼10 mm TL), was characterized by a high life expectancy, a decrease in mortality, and a notable maintenance of body size. This last observation is due to a slight reduction in the growth rate with a subsequent rebound in the curve, that having been also observed by Silva et al. (2008) in B. tenuissimus. During this period of time, energy is concentrated on body maintenance, growth and reproductive effort (Carvalho et al., 2009).

Díaz, Martin & Rumi (2023), using a logistic regression analysis, calculated gonadic maturity size of B. bonariensis at 12 mm total shell length. During this study, growth was followed and an age in days was established, according to the average TL of the shell. Thus, it can be inferred that gonadal maturity is reached after 200 days, when the individuals reach a length of 12 mm, at which point the growth rate decreases slightly. Therefore, newly-born individuals are not able to reproduce until the following year. The approximation of the size of the first reproduction is 16.68 mm (TL) on the average, at one year of life, when a second (lesser) decrease in the growth rate occurs. In B. tenuissimus Silva et al. (2008) reported that, between days 180 and 200 of the experiment (∼15–18 mm in total shell length), the growth slowed down, but did not stop after the first clutch, both in snails arranged in groups and in pairs. Therefore, in that species the size of the first reproduction is reached earlier than in B. bonariensis. In contrast, C. aspersum reaches sexual maturity at 13 weeks of age (Nicolai et al., 2010) and before one year of age in the case of L. fulica (Armiñana García, Fimia Duarte & Iannacone, 2020).

On the other hand, a very useful tool is the performance index φ (Pauly & Munro, 1984), which allows comparison of growth efficiency between species through the availability of parameters in different data sets. Thus, it was found that B. bonariensis has a lower growth efficiency than the three exotic species with which it was compared, L. fulica, C. aspersum and R. decollata. For this reason, the necessity to test different models is emphasized, as that approach not only reveals an aspect of the snail’s life history, but also enables other authors to make comparisons through the use of the aforementioned statistics.

Although A. gracile has a lower growth efficiency, this species has other characteristics such as an early sexual maturity (20–36 days), a prolonged reproductive period, and a short life expectancy (117.82 days), which combination facilitates its establishment in new habitats.

With respect to life history, Heller (1990) classifies mollusks in two categories: those with a short life span (less than two years, where he mentions the genus Bulimulus), and those with a long life span (living more than two years and reproducing in at least two seasons). In the present study, under laboratory conditions, it was possible to rear the snails for almost two years, which coincides with Heller (1990). Notwithstanding, on the basis of the maximum lengths recorded in the collection material and the life expectancy that was calculated (1,020 days = 2.79 years), we can adjust the life span of the species and conclude that specimens of B. bonariensis live between two and three years. Even Silva et al. (2008); Silva et al. (2013) argue that B. tenuissimus is a long-lived species with greater reproductive success in specimens that reach larger sizes. During the experiment carried out by those authors in 2008, the population was maintained until the last specimen died with an average TL of 20.6 mm; that length is virtually the same size as in B. bonariensis (present study) since during the same time the individuals reached an average length of 20.24 mm. In a second experiment, Silva et al. (2013) found that specimens of B. tenuissimus, kept in groups, reached a length of 22.8 mm at 990 days (2.71 years).

Furthermore, according to Karkach (2006), within mollusks, gastropods and species of the family Strombidae follow curves in models such as the exponential, the logistic, Richard, Gompertz, von Bertalanffy or Brody’s sigmoid (Brody, 1945) that are applicable to the type of growth determined. According to this author’s definition, in what is refered to as determinate growth, this process stops at sexual maturity or continues for some time thereafter. This is in contrast to indeterminate growth, which is defined as a continuation of growth after maturity is reached. This type of indeterminate growth was observed in B. tenuissimus by Silva et al. (2008). In view of the similarities of the latter species with B. bonariensis as well as the shell size calculated at gonadic maturity (Díaz, Martin & Rumi, 2023), we can conclude that B. bonariensis also possesses indeterminate growth, as do other terrestrial gastropod species such as Subulina octona (Dávila & de Almeida Bessa, 2005), Bradybaena similaris (Carvalho, Almeida Bessa & D’ávila, 2008), Habroconus semenlini (Silva et al., 2009), L. fulica (Armiñana García, Fimia Duarte & Iannacone, 2020), and A. gracile (Nandy & Aditya, 2022).

Upon this basis, we can reflect on the mathematical models that are being implemented at present and have been implemented for some time. Hernandez-Llamas & Ratkowsky (2004) expressed the need of reconsidering the use of growth models mainly in fish, crustaceans and mollusks, and the need to implement new models. Knowing the basic aspects of a life history, such as having determinate or indeterminate growth, we must begin to apply other growth curves that may be more explanatory, such as the multiphase and polynomial curves, mentioned by Karkach (2006) and which are specific to the indeterminate growth type.

For its part, B. bonariensis evidenced a type III survival curve, typical of r-strategists with a high mortality in the first month of life (Smith & Smith, 2007). This strategy was also observed by Rodríguez-Potrony et al. (2020) in Polymita brocheri, where only 12% of the population survived at the end of the breeding period. Staikou (1998) also reported a similar survival and a high mortality rate during the first year of life in Cepaea vindobonensis; and in Placostylus species, a high juvenile mortality of 60% was recorded (Brescia et al., 2008). Bradybaena similaris (Carvalho, Almeida Bessa & D’ávila, 2008) and C. aspersum (Daguzan et al., 1981) also have high mortalities in the early stages. In B. tenuissimus, mortality during the study period ranged from 35% in a group that was wetted daily (Silva et al., 2009), to 60% in a group reared on a combined diet (Meireles et al., 2008). In B. bonariensis, the monitoring was carried out until the last individual died. All of the above is in contrast to what has been reported for other terrestrial gastropods with a k-strategy, such as Leptinaria unilamellata (Carvalho et al., 2009), and which have exhibited a survival rate as high as 91.7% for Megalobulimus maximus (Rengifo Vásquez, Padilla Pérez & Mori Pinedo, 2004) and 85% for Powelliphanta augusta (James et al., 2013).

Comparative studies are essential for establishing evolutionary patterns within each group because differences in growth play a very important role in the evolution of species (Brey, 1999). Knowing, for example, the mortality rate, enables us to understand how this parameter shapes snail life history through investing more energy in age classes that contribute to reproduction, thus resulting in a longer life expectancy and more extensive reproductive period (Stearns, 2000). In addition, a wide distribution range impacts the biologic processes of organisms by affecting reproduction and survival, which capabilities translates into phenotypic variation within a species, so as to drive the evolution of biologic diversity (Gaitán-Espitia & Nespolo, 2014).

In South America, few studies have been undertaken in relation to life cycles, growth, and reproduction of Bulimulus species (Meireles et al., 2008; Meireles et al., 2010; Silva et al., 2008; Silva et al., 2009; Silva et al., 2013; present study). A continuing pursuit of studies with this approach in different terrestrial gastropod species is crucial not only because those species can become pests (Clemente et al., 2007) of sanitary and economic significance (Silva et al., 2013) or of commercial cultivation (Daguzan et al., 1981; Rengifo Vásquez, Padilla Pérez & Mori Pinedo, 2004), but also because of their relevance to ecology (Staikou, 1998), and conservation relevance (Brescia et al., 2008; James et al., 2013; Rodríguez-Potrony et al., 2020). In addition, invasive species may threaten the loss of native biodiversity (Carvalho da Silva & Omena, 2014).

Thus, during the development of this work we studied the life history of B. bonariensis in simulated conditions, which information could prove useful in future work as an element of comparison with quantitative data from a field populations.

Supplemental Information

Supplemental Information 1 Measurements of the total length of the shell of each specimen of Bulimulus bonariensis throughout its life

Click here for additional data file.

Supplemental Information 2 Protocol approved by CICUAE (FCNyM, UNLP)

Click here for additional data file.

We thank the directors of the SERByDE laboratory, Dra. Alejandra Rumi and Dra. Inés César, for providing the facilities and the necessary material for the study.

Additional Information and Declarations

Competing Interests

Author Contributions

Ethics

Data Availability

The authors declare there are no competing interests.

Ana Carolina Díaz conceived and designed the experiments, performed the experiments, analyzed the data, prepared figures and/or tables, authored or reviewed drafts of the article, and approved the final draft.

Stella Maris Martin conceived and designed the experiments, authored or reviewed drafts of the article, and approved the final draft.

The following information was supplied relating to ethical approvals (i.e., approving body and any reference numbers):

The experiment was performed in accordance with the recommendations from Comité Institucional para el Cuidado y Uso de Animales de Estudio (CICUAE)-Facultad de Ciencias Naturales y Museo-(FCNyM-UNLP)

The following information was supplied regarding data availability:

The raw measurements are available in the Supplementary File.

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
