# Peer review of "Numerical and biomass growth study of Bulimulus bonariensis (Rafinesque, 1833) (Gastropoda: Bulimulidae) under laboratory conditions"

_PeerJ, doi:10.7717/peerj.16803_

## Round 0.1 · original submission · Minor Revisions

I have received evaluations of your manuscript from two expert reviewers and their comments can be seen below and in an annotated document that is attached. Both reviewers agree that this manuscript has important information, but have also suggested some corrections and modifications that should be made to the document. I agree with their evaluations and suggest that you revise your manuscript ensuring that you follow the reviewers´ suggestions. Please ensure you include a detailed response to reviewers that clearly shows what changes have been made and where these changes can be found.

·

Basic reporting

no comment

Experimental design

no comment

Validity of the findings

no comment

Additional comments

no comment

·

Basic reporting

With regards to basic reporting, the article meets the requirements of PeerJ.

Experimental design

The design of the experiment is okay, however I think the findings would be stronger/more useful if this study had been performed with clutches from more than 10 adult snails. There is insufficient information to determine if all clutches came from 1 or maybe 2 adults as this species is hermaphroditic, or if there were multiple mating pairs from which clutches were taken. That information would be useful in interpreting results. Fortunately, the data are very consistent, but authors should still include that information as well as origin of the snails- were they field collected? Taken from laboratory colonies?

Validity of the findings

The authors were thorough in providing data and description of the analyses performed. The conclusions are clear based on data presented.

Additional comments

This study was performed with clutches from a small cohort, presumably at one time of year, however I wonder if these data would be different from clutches laid at different times of the year (this only really applies if adults were field collected).

The authors state that snails become reproductively viable at approximately 200 days of life, however no offpsring production was noted in this study. Were snails kept separate for the duration of the study? Did reproduction occur and did it have impacts on longevity of adults?

Lines 435-437 The authors mention another paper where the authors suggest the need to reconsider the use of growth models, but provide no logic for this statement. The sentence after ("In addition to the need to implement new models") is confusing- was that supposed to be part of the sentence before it or a partially completed thought?

Lines 471-471: This study did not describe population patterns. This was a straightforward life history study under simulated conditions. If you paired it with quantitative data from a field population, then you could likely use your data & analyses to explain those data.

---

## Round 0.2 · accepted · Accept

I am satisfied with the modifications that have been made to the manuscript and in my opinion it is ready for publication in PeerJ. Felices fiestas!